# TraCache: Trajectory-Aware Feature Prediction for Training-Free Diffusion Transformer Acceleration

## Abstract

Diffusion transformers have achieved remarkable success across various generative tasks but suffer from high inference costs. A promising line of work addresses this by reusing features across timesteps to minimize computational redundancy. However, existing methods degrade quality as temporal gaps increase due to *trajectory shifts in the feature space*. We propose ***TraCache***, a trajectory-aware caching framework that models feature evolution across timesteps. Instead of direct reuse, TraCache fits local feature trajectories and extrapolates accurate predictions, mitigating drift while maintaining quality under aggressive acceleration. Extensive experiments on image and video generation show that TraCache significantly outperforms prior cache-based methods, especially in high skip-rate regimes. For instance, TraCache accelerates PixArt-$\alpha$ by $3.86\times$ and Open-Sora by $3.74\times$, while on DiT-XL/2, it provides a $4.51\times$ acceleration with near-original visual fidelity.

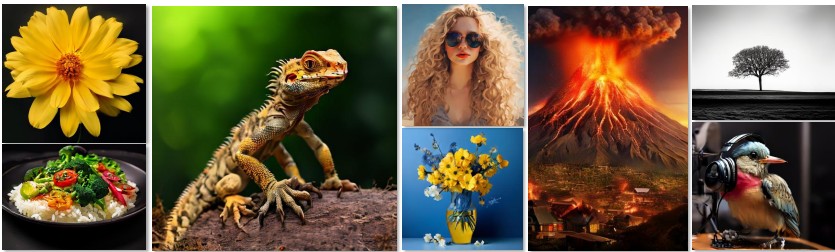

Figure 1: Accelerating PixArt-$\alpha$ by $3.86\times$ with 20 DPM-Solver++ steps.

## 1 Introduction

Diffusion models (Dhariwal & Nichol, 2021; Sohl-Dickstein et al., 2015) have emerged as a leading paradigm in generative modeling, achieving remarkable performance in image (Rombach et al., 2022), and video synthesis (Blattmann et al., 2023; Ma et al., 2024a; Zheng et al., 2024). Among various architectures, Diffusion Transformers(DiTs) (Peebles & Xie, 2023) exhibit superior expressiveness. However, the computational demands during inference, stemming from iterative denoising steps over deep models, remain a significant hurdle to widespread deployment.

Numerous acceleration methods have been explored (Shang et al., 2023; Fang et al., 2023; Ma et al., 2024c) to alleviate this burden. Among them, feature caching (Li et al., 2023a; Wimbauer et al., 2024; Ma et al., 2024b) has gained traction as a lightweight and effective technique. By exploiting the temporal similarity of intermediate representations, it stores features at selected timesteps and reuses them later, thereby avoiding redundant computations without modifying the base model. Existing studies primarily focus on "***when to cache***" and "***what to cache***": some employ training-based (Ma et al., 2024b; Shen et al., 2025) or training-free strategies (Liu et al., 2024a) to determine at which timesteps cached values should be used to skip computation, while others investigate mechanisms such as classifier-free guidance (CFG) caching (Lv et al., 2024) or token-level caching (Zou et al., 2024). Those approaches directly reuse cached values, relying heavily on the assumption of inter-timestep feature similarity. However, in practice this similarity degrades rapidly as the temporal gap increases, leading to pronounced artifacts and noticeable quality loss (Figure 2 (a-b)).

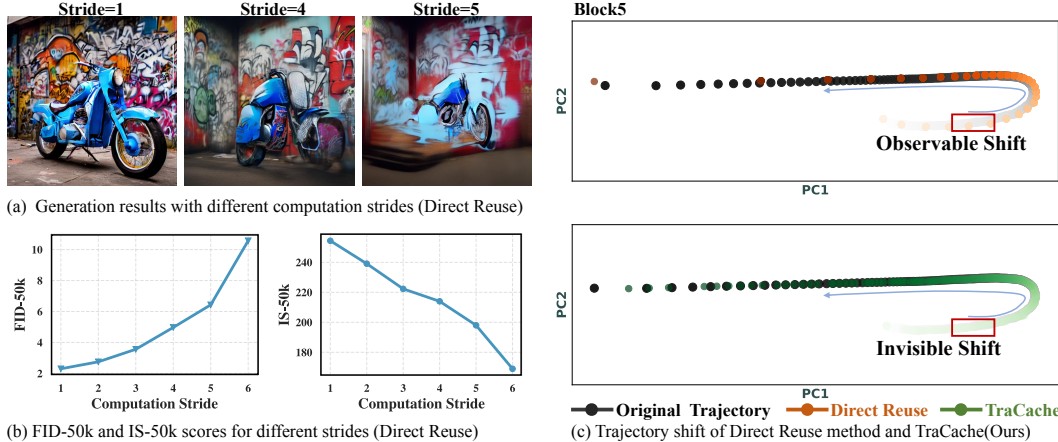

Figure 2: Under the **direct reuse** strategy, (a) generation quality significantly degrades as computation stride grows with prompt "A blue motorcycle is parked in a corner covered in graffiti". Stride=1 corresponds to full-step inference. (b) Quantitatively, larger strides result in higher FID-50k and lower IS-50k scores, reflecting degraded generation quality. (c) PCA visualization of features, with blue arrows indicating the generation time sequence.

To investigate the root cause of this quality degradation, we conducted an in-depth analysis of feature evolution across timesteps. By projecting per-timestep module outputs into a low-dimensional space via Principal Component Analysis (PCA) and visualizing them in temporal sequence order, shown in Figure 2 (c), we arrive at two key observations: (1) the module outputs form a stable trajectory along the temporal sequence, and (2) direct reuse causes a observable trajectory shift. Based on these findings, we propose a straightforward yet effective insight: *can we mathematically model and fit these trajectories to reduce deviation?* Specifically, we delve deeply into "***how to use cache***" and introduce TraCache, a trajectory-aware feature caching framework that leverages local historical information to fit and extrapolate the feature trajectory, thereby mitigating drift and maintaining high generation quality even under aggressive stride settings. This enables substantially higher acceleration ratios while preserving visual fidelity.

Extensive experiments across multiple generation tasks demonstrate that TraCache delivers significant acceleration with negligible quality degradation. By modeling feature trajectories rather than relying on direct reuse, our method produces significantly more accurate feature approximations that effectively mitigate trajectory deviation. This trajectory-aware approach enables TraCache to maintain high-fidelity generation even under aggressive acceleration settings where existing methods fail. Remarkably, TraCache attains up to **4.51×** speedup while maintaining excellent generation quality, whereas prior methods suffer significant performance drops.

To summarize, our main contribution is threefold:

- We provide a comprehensive analysis of feature evolution in diffusion process, revealing why direct reuse strategies fail at large temporal gaps and establishing the theoretical foundation for trajectory-aware approaches.
- We propose TraCache, a trajectory-aware feature caching framework that employs local polynomial fitting to predict future features based on historical information. This approach effectively mitigates drift while maintaining computational efficiency.
- We evaluate TraCache on diverse tasks, demonstrating state-of-the-art acceleration with minimal quality loss, consistently outperforming existing caching methods across speedup levels.

## 2 RELATED WORK

**DMs and the Applications.** Diffusion models(DMs) (Fei et al., 2023; Dhariwal & Nichol, 2021) have achieved significant success in generation tasks. The Denoising Diffusion Probabilistic Models (DDPMs) (Ho et al., 2020) generate higher-quality images than generative adversarial net-

works(GANs) (Li et al., 2020; Brock et al., 2018) via iterative denoising. To enhance sampling efficiency and enable high-resolution synthesis, Latent Diffusion Models(LDMs) (Rombach et al., 2022) operate within a compressed latent space. Early diffusion architectures primarily adopted U-Net backbones (Ronneberger et al., 2015; Podell et al., 2023), but their scalability is limited. To address this, Diffusion Transformers (DiTs) (Peebles & Xie, 2023) replace the U-Net with Transformer-based architecture, substantially boosting representational capacity and model scalability. This architectural evolution has facilitated advances in generation tasks, including high-fidelity image synthesis (Esser et al., 2024) and long-form video generation (Ma et al., 2024a; Kong et al., 2024). For instance, OpenAI's Sora (Brooks et al., 2024) showcases the strength of DiT-based architectures in capturing physical dynamics and long-range temporal dependencies.

**Training-free Acceleration of Diffusion Models.** The generation process in diffusion model is resource-intensive, primarily due to its iterative nature and the high per-step computational overhead. To mitigate this cost, prior work has explored two main strategies: reducing the number of inference steps and lowering the cost per step. The first category accelerate inference by reformulating the diffusion process. For instance, DDIM (Song et al., 2020) converts DDPM (Ho et al., 2020) into a non-Markovian process, enabling deterministic and faster sampling. The DPM-Solver series (Lu et al., 2022a;b) employs exponential integrators to efficiently solve the reverse-time ordinary differential equation(ODE), while other methods target more accurate solutions to stochastic differential equations(SDEs) (Dockhorn et al., 2021; Jolicoeur-Martineau et al., 2021) or ODEs (Zhang & Chen, 2022b; Zhang et al., 2022; Liu et al., 2022a). The second category targets the cost per step including model parameter sizes reduction (Li et al., 2023c; Kim et al., 2023; Castells et al., 2024; Zhang et al., 2024; Liu et al., 2024b), and the use of low-precision arithmetic (Li et al., 2023b; He et al., 2023). Moreover, dynamic inference approaches (Liu et al., 2023; Pan et al., 2024) have been proposed, which dynamically select models of varying capacities depending on the sampling stage.

**Feature Caching in Diffusion Models.** Feature caching accelerates diffusion models by reusing features in adjacent timesteps, effectively avoiding redundant computations. Methods such as DeepCache (Ma et al., 2024c) and FasterDiffusion (Li et al., 2023a) leverage this principle by, respectively, reusing high-level feature maps and encoder outputs within the U-Net architecture. In Transformer-based models, caching approaches can be broadly divided into dynamic and static paradigms. Dynamic methods adaptively decide whether to reuse cached features or recompute them at each timestep, while static methods reuse cached features at predetermined intervals. Among dynamic approaches, L2C (Ma et al., 2024b) and LazyDiT (Shen et al., 2025) learn policies for cache usage, whereas training-free methods such as TeaCache (Liu et al., 2024a) adopts a timestep-aware strategy, and AdaCache (Kahatapitiya et al., 2024) makes caching decisions based on the rate-of-change from previously computed features. For static methods, $\Delta$-DiT (Chen et al., 2024) stores residuals between block outputs, while FORA (Selvaraju et al., 2024) caches attention and MLP outputs. PAB (Zhao et al., 2024) employs different caching intervals for self-attention, cross-attention, and feed-forward network components. FasterCache (Lv et al., 2024) investigates redundancy in classifier-free guidance (CFG) (Ho & Salimans, 2022). ToCa (Zou et al., 2024) pushes caching granularity to the token level, although it becomes incompatible with efficient attention mechanisms (Dao et al., 2022) due to its reliance on attention scores. Dynamic methods primarily focus on "*when to cache*," while static methods concentrate on "*what to cache*," yet insufficient attention has been paid to "*how to use cache*". Current methods most reuse cached features directly, which fails under aggressive acceleration settings due to trajectory shift. Substantial opportunities remain to be explored.

## 3 METHOD

### 3.1 PRELIMINARY

**Diffusion Model.** Diffusion model generates samples from the complex data distribution by simulating the reversal of a forward noising process. The forward process progressively perturbs a data points $\mathbf{x}(0)$ over time via a SDE. This is mathematically formalized by:

$$\mathrm{d}\mathbf{x}(t) = f(t)\mathbf{x}(t)\mathrm{d}t + g(t)\mathrm{d}\mathbf{w}(t), \qquad (1)$$

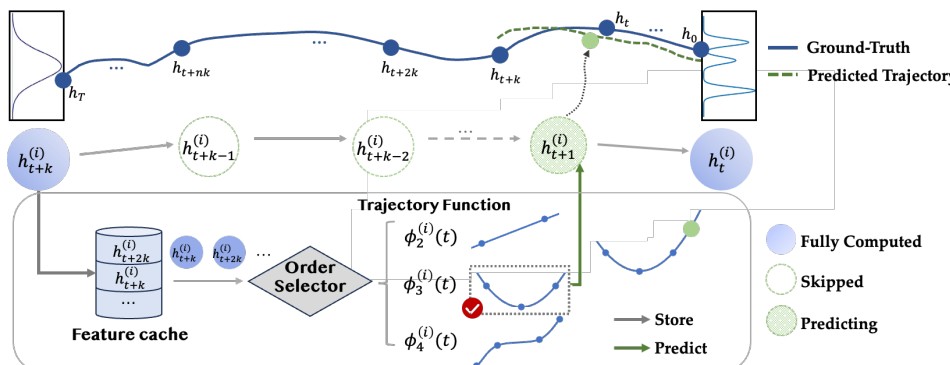

Figure 3: **Overview of the TraCache framework**. Computes and caches features every $k$ steps, using extrapolation for efficiency at other timesteps. An order selector adaptively manages prediction.

where $\mathbf{w}(t)$ is a standard Wiener process, and $f(t), g(t)$ are the drift and diffusion coefficients controlling the rate and scale of noise injection.

The reverse process reconstructs $\mathbf{x}(0)$ from a Gaussian noise by following a learned trajectory:

$$d\mathbf{x}(t) = [-f(t)\mathbf{x}(t) + g^2(t)\nabla_{\mathbf{x}(t)}\log p_t(\mathbf{x}(t))]dt + g(t)d\bar{\mathbf{w}}(t), \qquad (2)$$

where $\bar{\mathbf{w}}(t)$ signifies the reverse-time Wiener process, and $p_t(\mathbf{x}(t))$ is the data distribution at time $t$. And a neural network $s_\theta(\mathbf{x}_t, t)$ is trained to estimate the score function $\nabla_{\mathbf{x}(t)}\log p_t(\mathbf{x}(t))$.

In practice, the continuous process is discretized into a sequence of latent variables, denoted as $\{\mathbf{x}_T, \mathbf{x}_{T-1}, ..., \mathbf{x}_0\}$. Notably, due to the continuous nature of the underlying SDE, these latent states collectively trace out a smooth trajectory, which serves as the foundation for our method.

**Diffusion Transformer Architecture.** Diffusion Transformers (DiTs) consist of $L$ stacked blocks $\{f_1, \ldots, f_L\}$. At each timestep $t$, the denoising process on latent input $\mathbf{x}_t$ can be represented as a sequence of hidden state transformations:

$$\mathbf{x}_t \xrightarrow{f_1} \mathbf{h}_t^{(1)} \xrightarrow{f_2} \cdots \xrightarrow{f_L} \mathbf{h}_t^{(L)}, \qquad (3)$$

where $\mathbf{h}_t^{(i)} = f_i \circ \cdots \circ f_1(\mathbf{x}_t)$ denotes the output of the $i$-th Transformer block at timestep $t$ and the final output $\mathbf{h}_t^{(L)}$ serves as $\mathbf{x}_{t-1}$. This block-wise decomposition provides fine-grained access to intermediate representations, which TraCache leverages for trajectory-aware feature prediction.

**Vanilla Feature Caching.** Given a computation stride $k$, the intermediate hidden state computed at $t+k$ is directly reused at subsequent steps $\{t+(k-1), \ldots, t+1\}$, bypassing the computation of block $f_i$ at those steps. As stride $k$ increases, the drift between cached and actual features becomes increasingly pronounced, resulting in noticeable quality degradation.

To address this, we reinterpret feature caching from a mathematical modeling perspective: instead of directly reusing features that become increasingly dissimilar over time, we leverage historical trajectory information to extrapolate future feature representations.

## 3.2 TRAJECTORY APPROXIMATION IN LATENT SPACE

**From Observation to Mathematical Modeling.** Our PCA analysis in introduction section reveals two critical insights: (1) features form stable a trajectory across timesteps, and (2) direct reuse induces deviation. These observations suggest a fundamental opportunity: if features naturally follow predictable paths, we can mathematically model these trajectories rather than blindly reusing outdated representations. This motivates modeling intermediate representations $\mathbf{h}_t^{(i)}$ as discrete samples from an underlying continuous trajectory. Specifically, we posit the existence of a continuous function $\phi^{(i)} : \mathbb{R} \to \mathbb{R}^d$:

$$h_t^{(i)} = \phi^{(i)}(t) + \epsilon_t, \qquad (4)$$

where $\epsilon_t$ is minor perturbation. This reframes feature caching from assuming temporal similarity $\mathbf{h}_{t_1}^{(i)} \approx \mathbf{h}_{t_2}^{(i)}$ to predicting $\mathbf{h}_{t_2}^{(i)}$ via trajectory approximation using historical points.

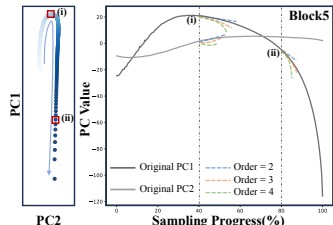 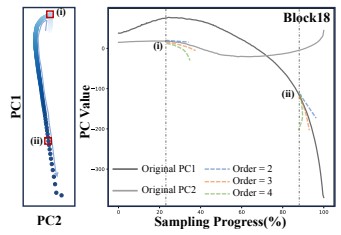 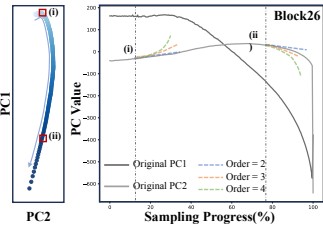

Figure 4: Visualization of trajectory fitting with different orders. In each subplot, the left panel shows the PCA features, and the right panel presents the fitted curves. The three subplots correspond to the results for blocks 5, 18, and 26, respectively. (i) Dynamic regions; (ii) Smooth regions.

**Proof of Trajectory Predictability.** The predictability of feature trajectories $\phi^{(i)}(t)$ stems from the inherent smoothness of diffusion processes. We establish this property through following analysis: (1) the latent variables $\mathbf{x}_t$ evolve smoothly due to the reverse SDE; (2) each intermediate representation is computed as $\mathbf{h}_t^{(i)} = f_i \circ \cdots \circ f_1(\mathbf{x}_t)$; (3) by applying the chain rule of differentiation, we obtain $\frac{d\mathbf{h}_t^{(i)}}{dt} = \left( \prod_{j=1}^i \frac{\partial f_j}{\partial \mathbf{h}_t^{(j-1)}} \right) \cdot \frac{\partial \mathbf{x}_t}{\partial t}$; (4) since each $f_j$ consists of differentiable components (LayerNorm, Attention, and MLP) and $\mathbf{x}_t$ varies smoothly, the composite function $\phi^{(i)}(t)$ inherits this smoothness. By Taylor's theorem (Turnbull, 1930), smooth functions admit local polynomial approximations with bounded approximation error, thus justifying trajectory-based prediction approach. A formal argument and visualization results provided in Appendix D.1.2.

### 3.3 TraCache: Trajectory-Aware Caching Algorithm

We present TraCache, a trajectory-aware feature caching framework to enhance sampling efficiency of diffusion models, particularly under large timestep skipping. Our core idea utilizes the inherent predictability of the latent trajectory (as justified in Section 3.2) by maintaining temporal history and fitting local trajectory for extrapolation-based prediction. This reduces the number of expensive model forward passes, offering significant speedups with minimal degradation in sample quality.

**Linear Prediction.** We begin with first-order approximation under local linearity assumptions. Given two last computed hidden states, $\mathbf{h}_{t_0}^{(i)}$ and $\mathbf{h}_{t_1}^{(i)}$ at timesteps $t_0 > t_1$, the modeled trajectory function is given by:

$$\phi^{(i)}(t) = \alpha \cdot \mathbf{h}_{t_0}^{(i)} + (1 - \alpha) \cdot \mathbf{h}_{t_1}^{(i)}, \quad \text{where } \alpha = \frac{t - t_1}{t_0 - t_1}. \tag{5}$$

Then we use this function to estimate $\hat{\mathbf{h}}_t^{(i)} = \phi^{(i)}(t)$, which provides a temporally aligned approximation with lower error compared to direct cache reuse.

**Higher-Order Prediction.** Linear interpolation works well for near-linear dynamics, but it struggle with nonlinear trajectory segments, especially under aggressive stride settings. To address this, we generalize our approach to higher-order polynomial interpolation using the Lagrange formulation. Using $n+1$ fully computed hidden states $(t_k, \mathbf{h}_{t_k}^{(i)})_{k=0}^n$, we define the trajectory function as:

$$\phi_n^{(i)}(t) = \sum_{k=0}^n \mathbf{h}_{t_k}^{(i)} \cdot \ell_k(t), \quad \text{where } \ell_k(t) = \prod_{\substack{q=0 \\ q \neq k}}^n \frac{t - t_q}{t_k - t_q}. \tag{6}$$

This polynomial predictor $\phi_n^{(i)}(t)$ matches each cached state at their corresponding timestep, i.e., $\phi_n^{(i)}(t_k) = \mathbf{h}_{t_k}^{(i)}$ for all $k$. By capturing nonlinear trajectory curvature, the high-order approximation offers a more accurate prediction. Furthermore, since the target timestep $t$ is known in advance, the interpolation weights $\alpha_k = \ell_k(t)$ can be precomputed once and reuse them for efficiency.

**Dynamic Order Selection.** Diffusion trajectories exhibit heterogeneous local characteristics, containing segments with gradual changes and rapid dynamics near inflection points. Fixed interpolation orders are suboptimal as different regions require different modeling complexity. Smooth

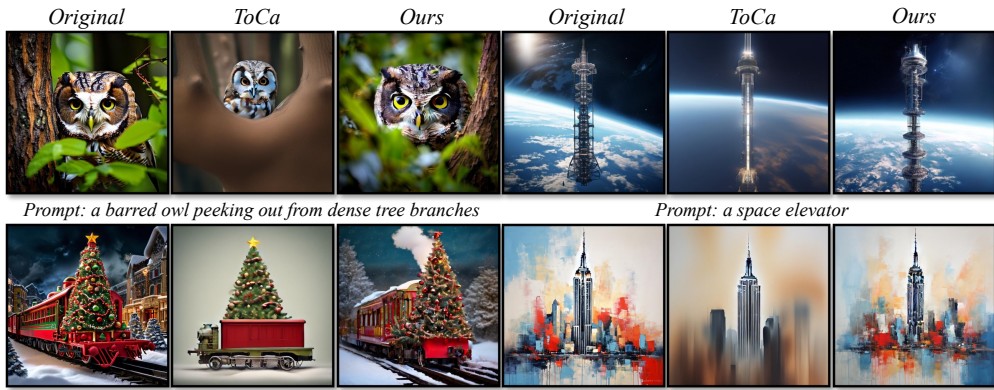

*Prompt: a barred owl peeking out from dense tree branches*      *Prompt: a space elevator*

*Prompt: a Christmas tree on a toy train*      *Prompt: an abstract painting of the Empire State Building*

Figure 5: Qualitative results for text-to-image generation on PartiPrompts using PixArt-$\alpha$ (512×512). We compare ToCa (2.61× speedup) and our method TraCache (2.74× speedup).

segments benefit from higher-order interpolation for more accurate estimation, while rapidly changing regions near inflection points require lower orders for stability. Figure 4 illustrates this: (1) a dynamic region favors low order, and (2) a smooth region is better captured with higher order.

We propose an adaptive order selection mechanism and define the residual between successive polynomial orders as $\epsilon_n^{(i)}(t) = \|\phi_n^{(i)}(t) - \phi_{n-1}^{(i)}(t)\|$, where $\phi_n^{(i)}(t)$ denotes the $n$-th order interpolated prediction at timestep $t$. $\epsilon_n^{(i)}(t)$ represents the difference between two predicted feature maps, thus its standard deviation measures whether changes are consistent across all feature dimensions. We select the order that minimizes this variance:

$$n^{*(i)}_t = \arg\min_n \mathrm{std}(\epsilon_n^{(i)}(t)). \tag{7}$$

Beyond the theoretical motivation, we analyze the relationship between the proposed std metric and the actual trajectory deviation. (See in Section 4.3) Further discussions on the choice of criterion can be found in Appendix D.1.4.

**In summary**, by formulating feature estimation as a trajectory interpolation problem, TraCache offers a principled way to reuse historical hidden states, allowing the model to bypass redundant computation during inference. Through linear and higher-order strategies, TraCache enables efficient, interpretable, and quality-preserving acceleration for diffusion-based generation.

## 4 EXPERIMENT

### 4.1 EXPERIMENTAL SETUP

**Models and Baselines.** We conduct experiments on three representative tasks: text-to-image generation with PixArt-$\alpha$ (Chen et al., 2023), text-to-video generation with Open-Sora 1.2 (Zheng et al., 2024), and class-conditional image generation with DiT/XL-2 (Peebles & Xie, 2023). For each model, we adopt its standard sampling scheduler: DPM-Solver++ (Lu et al., 2022b) with 20 steps for PixArt-$\alpha$, rflow (Liu et al., 2022b) with 30 steps for Open-Sora, and DDIM (Song et al., 2020) with 50 steps for DiT/XL-2. We compare TraCache against recent acceleration techniques, including advanced samplers (Zhang & Chen, 2022a), $\Delta$-DiT (Chen et al., 2024), FORA (Selvaraju et al., 2024), and ToCa (Zou et al., 2024). For video generation task, we further include methods specifically tailored to video scenarios, including PAB (Zhao et al., 2024), AdaCache (Kahatapitiya et al., 2024), FasterCache (Lv et al., 2024), and TeaCache (Liu et al., 2024a). For class-conditional image generation, we additionally consider training-based methods such as L2C (Ma et al., 2024b) and LazyDiT (Shen et al., 2025). Full configuration details are provided in the Appendix D.2.

**Evaluation Metrics.** We evaluate our method from two perspectives: generation efficiency and quality. For efficiency, we report Multiply-Accumulate Operations (MACs) and inference latency. For generation quality, we adopt task-specific metrics. In text-to-image generation, we use 5,000 MS-COCO 2017 validation set (Lin et al., 2014) and 1,630 PartiPrompts (Yu et al., 2022) as prompts, generating images at resolution of 256×256, and 512×512. Outputs are evaluated using CLIP Score

Table 1: **Quantitative comparison in text-to-image** on MS-COCO2017 and PartiPrompts using PixArt-$\alpha$ with 20 DPM++ steps (default). FORA[1–3] and ToCa[1–2] denote variants with distinct configurations as described in original works. TraCache$_{k,o}$ uses stride $k$ and a max interpolation order $o$. 'Res.' indicates image resolution. Methods are grouped by comparable acceleration ratios.

| Res. | Methods | Efficient Attention | Efficiency | | | | MS-COCO2017 | PartiPrompts |
|---|---|---|---|---|---|---|---|---|
| | | | Latency(s)↓ | Speed↑ | MACs(T)↓ | Speed↑ | CLIP↑ | CLIP↑ |
| | PixArt-$\alpha$ | ✓ | 0.574 | 1.00× | 5.68 | 1.00× | 31.18 | 30.56 |
| 256×256 | DPM++10steps | ✓ | 0.288 | 1.99× | 2.84 | 2.00× | 30.43 | 29.74 |
| | Δ-DiT | ✓ | 0.428 | 1.34× | 3.74 | 1.52× | 30.24 | 29.52 |
| | FORA[1] | ✓ | 0.301 | 1.93× | 2.88 | 1.97× | 29.99 | 29.46 |
| | **TraCache$_{2,2}$** | ✓ | 0.315 | 1.82× | 2.91 | 1.95× | **31.09** | **30.51** |
| | DPM++7steps | ✓ | 0.205 | 2.80× | 1.99 | 2.86× | 28.88 | 28.37 |
| | FORA[2] | ✓ | 0.207 | 2.77× | 2.02 | 2.81× | 28.65 | 28.22 |
| | ToCa[1] | ✗ | 0.388 | 1.48× | 2.19 | 2.59× | 29.04 | 28.73 |
| | **TraCache$_{3,2}$** | ✓ | 0.229 | 2.51× | 2.06 | 2.76× | **30.84** | **30.12** |
| | ToCa[2] | ✗ | 0.246 | 2.33× | 1.51 | 3.76× | 28.66 | 28.01 |
| | **TraCache$_{4,2}$** | ✓ | 0.161 | 3.56× | 1.48 | 3.85× | **29.93** | **29.71** |
| 512×512 | DPM++20steps | ✓ | 0.859 | 1.00× | 21.67 | 1.00× | 31.67 | 30.78 |
| | DPM++10steps | ✓ | 0.427 | 2.01× | 10.84 | 2.00× | 30.75 | 30.31 |
| | FORA[1] | ✓ | 0.450 | 1.91× | 10.94 | 1.98× | 30.46 | 30.29 |
| | **TraCache$_{2,2}$** | ✓ | 0.480 | 1.79× | 11.17 | 1.94× | **31.47** | **30.66** |
| | ToCa[1] | ✗ | 0.577 | 1.49× | 8.30 | 2.61× | 28.59 | 28.30 |
| | **TraCache$_{3,2}$** | ✓ | 0.348 | 2.47× | 7.91 | 2.74× | **30.97** | **30.43** |
| | FORA[3] | ✓ | 0.221 | 3.89× | 5.46 | 3.97× | 27.63 | 27.24 |
| | ToCa[2] | ✓ | 0.364 | 2.36× | 5.73 | 3.78× | 28.24 | 27.98 |
| | **TraCache$_{4,2}$** | ✓ | 0.250 | 3.44× | 5.61 | 3.86× | **29.88** | **29.56** |

(on ViT-G/14 model) (Hessel et al., 2021). In text-to-video generation, we adopt the VBench (Huang et al., 2024) benchmark and generate 5 videos per prompt for all 950 prompts. Each video consists of 51 frames at 480p resolution with a 9:16 aspect ratio. For class-conditional image generation, we synthesize 50,000 images across 1,000 ImageNet (Deng et al., 2009) classes at 256×256 resolution, and assess quality using FID (Nash et al., 2021), sFID (Nash et al., 2021), and Inception Score (IS) (Salimans et al., 2016).

**Implementation Details.** All experiments are conducted on NVIDIA A800 80GB GPUs with FlashAttention (Dao et al., 2022) enabled by default. However, since ToCa requires computation of attention scores, it cannot use any efficient-attention implementations. Unless otherwise specified, we denote the computation stride by $k$ (performing computation every $k$ steps) and the maximum interpolation order by $o$.

## 4.2 MAIN RESULTS

**Text-to-Image Generation.** Table 1 compares text-to-image performance of PixArt-$\alpha$ with various acceleration methods. Across both 256×256 and 512×512 resolutions, TraCache variants consistently achieve comparable or superior CLIP scores to baselines while providing significant latency and MAC reductions. Higher strides (e.g., TraCache$_{4,2}$) deliver the largest speed-ups(3.86×) with only minor quality loss, demonstrating the scalability of our approach.

Qualitative results in Figure 5 reveal that TraCache preserves visual fidelity and semantic alignment. In contrast, ToCa exhibits blurry textures and large color artifacts. This demonstrates that methods relying on direct reuse (e.g., ToCa) suffer from severe quality deterioration, whereas TraCache's trajectory-aware prediction bridges feature gaps and preserves generation quality.

**Text-to-Video Generation.** As illustrated in Table 2, TraCache delivers strong performance in text-to-video generation. TraCache$_{2,2}$ achieves notable efficiency gains with minimal quality loss. Even at its most aggressive setting, TraCache$_{4,2}$ attains the highest acceleration (3.4× latency, 3.7× MACs) with only a slight drop in VBench performance, demonstrating the scalability and robustness of our approach under aggressive acceleration. Figure 6 shows that TeaCache misses small structures (e.g., a building) and distorts the Eiffel Tower, while our method preserves both. In another example,

Table 2: **Quantitative comparison of text-to-video generation** for Open-Sora on VBench with 30 rflow steps (default). TeaCache[1–2] denote variants with distinct configurations as described in original works. *Results are from TeaCache. TraCache$_{k,o}$ uses stride $k$ and a max interpolation order $o$. Methods are grouped according to comparable acceleration ratios.

| Method | Latency(s)↓ | Speed↑ | MACs(P)↓ | Speed↑ | VBench(%)↑ |
|---|---|---|---|---|---|
| Open-Sora | 72.58 | 1.00× | 1.66 | 1.00× | 78.85 |
| rflow-15steps | 37.61 | 1.93× | 0.83 | 2.00× | 76.69 |
| Δ-DiT* | - | - | 1.63 | 1.02× | 78.21 |
| FORA | 38.20 | 1.90× | 0.85 | 1.95× | 76.83 |
| PAB | 53.76 | 1.35× | 1.32 | 1.26× | 76.56 |
| FasterCache | 45.22 | 1.61× | 0.99 | 1.68× | 78.28 |
| TeaCache[1] | 36.47 | 1.99× | 0.86 | 1.93× | 78.11 |
| **TraCache$_{2,2}$** | 41.47 | 1.75× | 0.84 | 1.98× | **78.64** |
| rflow-10steps | 24.85 | 2.92× | 0.55 | 3.00× | 74.54 |
| TeaCache[2] | 24.03 | 3.02× | 0.56 | 2.96× | 77.24 |
| AdaCache | 30.45 | 2.38× | 0.69 | 2.41× | 77.19 |
| **TraCache$_{3,2}$** | 25.83 | 2.81× | 0.55 | 3.00× | **78.01** |
| **TraCache$_{4,2}$** | 21.22 | 3.42× | 0.44 | **3.74×** | 77.38 |

Table 3: **Quantitative comparison of class-conditional** on ImageNet256×256 with DiT-XL/2 and 50 DDIM sampling steps by default. ToCa[1–2] denote variants using distinct configurations as described in the original works. TraCache$_{k,o}$ uses stride $k$ and a max interpolation order $o$. Methods are grouped by comparable acceleration ratios.

| Method | Efficient Attention | Extra Training | Efficiency | | | | FID↓ | sFID↓ | IS↑ |
|---|---|---|---|---|---|---|---|---|---|
| | | | Latency(s)↓ | Speed↑ | MACs(T)↓ | Speed↑ | | | |
| DiT-XL/2 | ✓ | ✗ | 0.495 | 1.00× | 11.44 | 1.00× | 2.30 | 4.33 | 241.24 |
| DDIM-20steps | ✓ | ✗ | 0.202 | 2.45× | 4.58 | 2.50× | 3.59 | 4.98 | 221.90 |
| DEIS-20steps | ✓ | ✗ | 0.205 | 2.41× | 4.58 | 2.50× | 2.43 | 5.58 | 233.33 |
| FORA | ✓ | ✗ | 0.183 | 2.71× | 4.12 | 2.78× | 3.53 | 6.38 | 227.78 |
| ToCa[1] | ✗ | ✗ | 0.367 | 1.35× | 4.92 | 2.33× | 2.88 | 4.74 | 234.15 |
| **TraCache$_{3,3}$** | ✓ | ✗ | 0.203 | 2.44× | 4.19 | 2.73× | **2.34** | **4.54** | **237.51** |
| ToCa[2] | ✗ | ✗ | 0.233 | 2.12× | 3.28 | 3.49× | 6.25 | 7.18 | 198.99 |
| L2C | ✓ | ✓ | 0.156 | 3.18× | 3.56 | 3.21× | 3.49 | 4.68 | 229.68 |
| LazyDiT | ✓ | ✓ | 0.146 | 3.39× | 3.20 | 3.57× | 4.39 | 5.57 | 212.33 |
| **TraCache$_{4,4}$** | ✓ | ✗ | 0.154 | 3.21× | 3.22 | 3.55× | **2.53** | 4.70 | **234.69** |
| **TraCache$_{5,4}$** | ✓ | ✗ | 0.120 | 4.11× | 2.54 | **4.51×** | 2.69 | **4.63** | 230.67 |

TeaCache produces blurry fountain streams and color shifts, whereas our method maintains fine details and accurate colors.

**Class-conditional Image Generation.** Table 3 compares TraCache with prior acceleration methods on DiT-XL/2. Without any extra training, TraCache achieves substantial latency and MAC reductions while maintaining or improving generation quality. TraCache$_{3,3}$ matches the speed of DDIM and DEIS but attains the best FID and sFID among all non-retrained baselines. TraCache$_{4,4}$ and TraCache$_{5,4}$ further increase acceleration (up to 4.5×) with only marginal quality degradation, outperforming methods such as ToCa and LazyDiT in both efficiency and fidelity. These results highlight TraCache's ability to scale to high acceleration while preserving strong visual quality without additional training. To demonstrate the independence of our method from sampler selection, we also conducted experiments on the DDPM sampler. The results are shown in the Appendix D.3.

**Memory Analysis.** We compare the peak GPU memory usage of our method against baseline approaches, as summarized in Table 4. While our method stores historical outputs, the additional memory cost relative to other caching methods remains modest. This is primarily because we cache at a coarser granularity: whereas ToCa store the attention and MLP outputs of the previous timestep for each block, our approach caches the entire block output. Importantly, this choice of caching granularity does not fundamentally affect performance; the key performance gains arise from the design of our TraCache mechanism. Detailed experimental results supporting this claim are provided in the Appendix D.1.5.

Table 4: Peak GPU VRAM.

| Method | VRAM (GB) |
|---|---|
| DiT-XL/2 | 2.89 |
| ToCa | 3.24 |
| TraCache$_{3,3}$ | 3.78 |
| TraCache$_{4,4}$ | 3.64 |
| TraCache$_{5,4}$ | 3.45 |

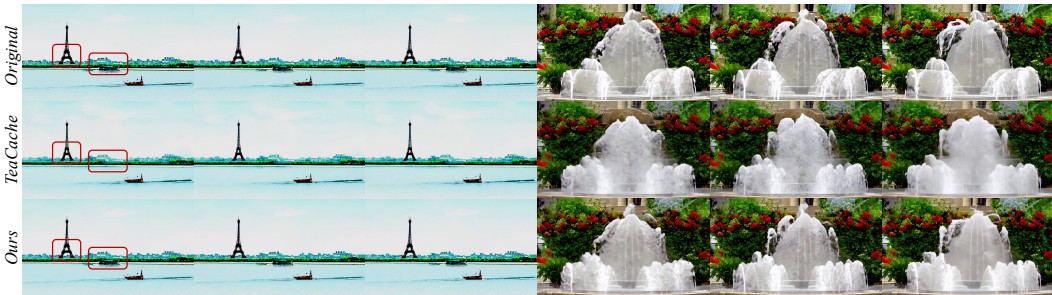

Figure 6: **Qualitative results for text-to-video generation** using Open-Sora. We compare TeaCache[1] (2.96× speedup) and our method (3.74× speedup).

### 4.3 ABLATION STUDIES

To analyze the contributions of each component in TraCache, we conduct ablation studies on DiT-XL/2 (Peebles & Xie, 2023).

**Effectiveness of Order Selection Metric.** For each interpolation order $n$, we compute (1) selection metric: the standard deviation (std) of $\epsilon_n^{(i)}(t)$, (2) trajectory shift: the principle component (PC) distance between $\phi_n^{(i)}(t)$ and ground-truth value in dynamic and smooth regions. When $n = 1$, the interpolation degenerates to directly using the last cached feature. As shown in Figure 7, both curves follow the same trend: the order minimizing std also minimizes the shift, indicating that this metric effectively

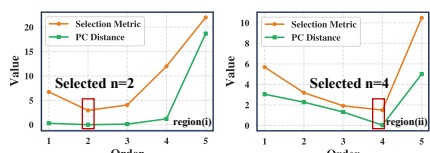

Figure 7: Metric and trajectory shift across interpolation orders. Left: dynamic regions; Right: smooth regions.

guides the selection of the best interpolation order. Moreover, it shows that dynamic regions tend to favor lower-order interpolation, while smooth regions favor higher-order interpolation.

**Influence of Interpolation Order.** Figure 8 (solid lines) shows that under a fixed-order setting, increasing the interpolation order initially improves generation quality but eventually causes a sharp decline, indicating that high-order interpolation can produce inaccurate fits. With dynamic order selection (dashed lines), quality improvements gradually saturate instead of deteriorating, as the mechanism detects when higher orders introduce large errors and selectively retains lower orders.

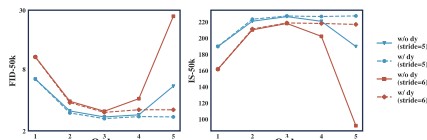

Figure 8: Ablation on interpolation order and dynamic selection. Left: FID; Right: IS.

**Effectiveness of the Dynamic Mechanism.** Figure 8 further shows that dashed curves (with the dynamic mechanism) consistently outperform solid curves (without it), particularly at higher orders. This confirms that the dynamic mechanism both selects the optimal interpolation order in low-order regions and mitigates numerical instability at high orders.

## 5 CONCLUSION

In this paper, we conduct an in-depth analysis of intermediate feature evolution in diffusion transformers and identify that the degradation of generation quality at high intervals in existing direct-reuse methods stems from trajectory shifts. Building on this observation, we propose TraCache, a trajectory-aware feature caching framework that fits and extrapolates feature trajectories to guide reuse during the denoising process. By leveraging local historical information, TraCache mitigates trajectory drift and maintains high generation quality, even under aggressive acceleration settings. Extensive experiments on both image and video generation tasks demonstrate that TraCache consistently outperforms prior cache-based methods, achieving substantial inference speedups while preserving visual fidelity.

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

## A  THE USE OF LARGE LANGUAGE MODELS

Large Language Models (LLMs) were only used to assist with minor language polishing and formatting. No LLMs were used for data analysis, experiment design, or generation of original content. All conceptualization, modeling, implementation, and evaluation were performed by the authors.

## B  ETHICS STATEMENT

This work does not involve human subjects, personally identifiable information, or sensitive data. No proprietary or non-public datasets are used. The proposed methods and findings do not present foreseeable harmful insights, applications, or security/privacy risks. There are no conflicts of interest, sponsorships, or legal/ethical concerns associated with this study.

## C  REPRODUCIBILITY STATEMENT

We have taken several steps to ensure the reproducibility of our results. The experimental settings and hyperparameter configurations are described in detail in Section 4.1 and Appendix D.2. Complete mathematical proofs and visualizations are provided in Appendix D.1.2. The source code will be submitted as supplementary material and released publicly upon acceptance. All datasets used in our experiments are publicly available, with data processing procedures specified in Section 4.1.

# D APPENDIX

## D.1 MORE DISCUSSION

### D.1.1 ANALYSIS OF FAILURE UNDER EXCESSIVE INTERVAL SIZE

To analyze the impact of large timestep intervals on feature reuse, we first apply PCA to visualize the latent trajectories. As shown in Figure 2(c), when the interval size is large, the feature trajectories deviate noticeably from their original paths. To quantify this effect, we compute the feature MSE between specific steps and their corresponding steps at different intervals (e.g., at interval $= 4$, we compute $\text{MSE}(\text{step}44, \text{step}48)$; at interval $= 5$, we compute $\text{MSE}(\text{step}44, \text{step}49)$). Using DiT-XL/2 with DDIM50 sampling, Table 5 shows that the MSE values are consistently high and increase further under larger intervals. This indicates that directly reusing features at large intervals leads to substantial errors.

| interval | step=44 | step=28 | step=6 |
|:---:|:---:|:---:|:---:|
| 4 | 13.608 | 482.391 | 621.62 |
| 5 | 16.078 | 614.421 | 817.79 |

Table 5: MSE at different intervals

### D.1.2 PROOF OF SMOOTHNESS IN TRANSFORMER BLOCKS W.R.T. TIMESTEP

We justify that a standard Transformer block defines a smooth function w.r.t. its input $x \in \mathbb{R}^d$.

We rely on the following principles:

- Sums, products, and compositions of smooth functions are smooth.
- Linear/affine transformations are smooth.
- Polynomial functions are smooth.
- The exponential function ($e^x$) is smooth.
- The square root function ($\sqrt{x}$) is smooth for $x > 0$.
- Division $f(x)/g(x)$ is smooth if $f(x)$ and $g(x)$ are smooth and $g(x) \neq 0$.

Each Transformer block consists of the following sequence:

$$\text{LayerNorm} \rightarrow \text{Attention} \rightarrow \text{ResidualAdd} \rightarrow \text{LayerNorm} \rightarrow \text{MLP} \rightarrow \text{ResidualAdd} \quad (8)$$

We now analyze the smoothness of each module.

**LayerNorm.** The LayerNorm operation is defined as:

$$\text{LN}(x) = \frac{x - \mu(x)}{\sigma(x)} \cdot \gamma + \beta \quad (9)$$

where $\mu(x)$ and $\sigma(x)$ are the mean and standard deviation value of $x$ respectively. $\sigma(x) = \sqrt{\sigma^2(x) + \epsilon_{var}}$ with a small $\epsilon_{var} > 0$ added for numerial stability. $\gamma$ and $\beta$ are learnable element-wise scale and shift parameters.

We examine each part:

- **Mean Calculation** ($\mu(x) = \frac{1}{D}\sum_{i=1}^{D} x_i$): a linear combination of $x_i$, hence smooth.
- **Variance Calculation** ($\text{Var}(x) = \frac{1}{D}\sum_{i=1}^{D}(x_i - \mu(x))^2$): a sum of smooth squared terms, hence smooth.
- **Standard Deviation Calculation** ($\sigma(x) = \sqrt{\text{Var}(x) + \epsilon_{var}}$): smooth since square root is smooth over $x > 0$, and $\epsilon_{\text{var}} > 0$ ensures positivity.

- **Normalization Step** ($z_i = \frac{x_i - \mu(x)}{\sigma(x)}$): a quotient of smooth functions with positive denominator, hence smooth.
- **Scaling and Shifting** ($\text{LN}(x)_i = z_i \cdot \gamma_i + \beta_i$): an affine transformation, hence smooth.

Therefore, $\text{LayerNorm}$ is a smooth function of $x$.

**Attention.** The scaled Dot-Product Attention is given by:

$$\text{Attn}(Q, K, V) = \text{softmax}(\frac{QK^T}{\sqrt{d_k}})V \tag{10}$$

Each component is analyzed as follows:

- **Q/K/V Projections** (e.g., $Q = XW_Q$): affine transformations of input $X$, hence smooth.
- **Scaled Dot-Product Scores** ($S = \frac{QK^T}{\sqrt{d_k}}$): matrix multiplication and scaling, both smooth operations.
- **Softmax Function** ($\text{softmax}(S)_i = \frac{e^{s_i}}{\sum_j e^{s_j}}$): smooth as it involves exponential and division of strictly positive terms.
- **Final Output** ($\text{Output} = \text{softmax}(S)V$): matrix product of smooth terms.

Therefore, $\text{Attention}$ is smooth with respect to $x$.

**MLP.** The typical form of an MLP layer in Transformer block is:

$$\text{MLP}(x) = W_2 \cdot \text{GELU}(W_1 x + b_1) + b_2 \tag{11}$$

Analyzing each part:

- **Inner Affine Transformation** ($L_1(x) = W_1 x + b_1$): affine transformation, hence smooth.
- **GELU Activation Function** ($\text{GELU}(y)$ where $y = L_1(x)$): The Gaussian Error Linear Unit is often defined as $\text{GELU}(y) = y \cdot \Phi(y)$, where $\Phi(y) = \frac{1}{2}[1 + \frac{2}{\sqrt{\pi}} \int_0^{\frac{y}{\sqrt{2}}} e^{-t^2} \mathrm{d}t]$. The integral of a smooth function with respect to its upper limit is smooth. Other part is composed of affine, linear transformation, and product, hence smooth.
- **Outer Affine Transformation** ($L_2(z) = W_2 z + b_2$ where $z = \text{GELU}(y)$): affine transformation, hence smooth.

Therefore, $\text{MLP}$ is a smooth function of $x$.

**Residual Connections.** The residual addition can be written as $y = x + F(x)$, where $F$ is one of the above modules. Since the sum of smooth functions is smooth, residual connections preserve smoothness.

**Conclusion.** Since all components in the Transformer block are smooth functions of the input $x$, their composition—the entire block—is also smooth in $x$.

To validate this empirically, we project attention, MLP outputs across timesteps into low-dimensional spaces using PCA. As shown in Figure 9 (b-c), the projected trajectories exhibit coherent and low-frequency motion, supporting the assumption of the Transformer's hidden representations.

### D.1.3 ABLATION FIXED-ORDER VS. DYNAMIC VARIANTS

We approach the order selection problem from the perspective of prediction error. While higher-order interpolation generally improves fitting, it also risks overfitting, which results in large prediction errors. Thus, Our selection principle is to increase the interpolation order as much as possible

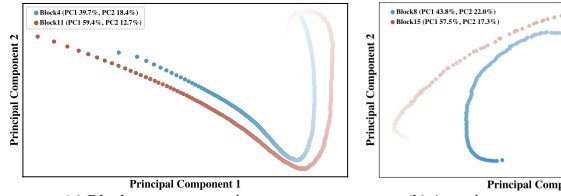 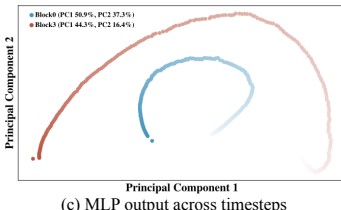

(a) Block output across timesteps     (b) Attention output across timesteps     (c) MLP output across timesteps

Figure 9: PCA visualization of block(a), attention(b), and MLP(c) outputs in DiT. The block outputs form smooth and consistent trajectories over time, suggesting that future features can be effectively predicted from past features. The stability of attention and MLP outputs supports the hypothesis in Section 3.2. The color of each point darkens along the sampling progression.

| Setting | FID ↓ | sFID ↓ | IS ↑ |
|---|---|---|---|
| k=5,o=2(dy) | 3.04 | 4.62 | 223.06 |
| k=5,o=2(fixed) | 3.15 | 4.71 | 220 |
| k=5,o=3(dy) | 2.65 | 4.65 | 227.67 |
| k=5,o=3(fixed) | 2.77 | 4.78 | 226.52 |
| k=5,o=4(dy) | 2.69 | 4.63 | 230.67 |
| k=5,o=4(fixed) | 2.89 | 4.99 | 220.94 |
| k=5,o=5(dy) | 2.75 | 4.63 | 227.57 |
| k=5,o=5(fixed) | 5.48 | 7.92 | 189.80 |
| k=6,o=3(dy) | 3.02 | 4.84 | 219.31 |
| k=6,o=3(fixed) | 3.22 | 5.01 | 217.67 |
| k=6,o=4(dy) | 3.23 | 5.09 | 218.27 |
| k=6,o=4(fixed) | 4.14 | 5.45 | 202.40 |
| k=6,o=5(dy) | 3.23 | 5.22 | 217.08 |
| k=6,o=5(fixed) | 26.33 | 23.68 | 92.130 |

Table 6: Ablation fixed-order vs. dynamic variants on DiT-XL/2 model with DDIM-50

without introducing overfitting. Ablation results comparing fixed-order and dynamic variants are presented in Table 6.

(1) Experiments conducted on DiT-XL/2 model with DDIM-50 sampler (Table1), Pixart- with DPM++20 (Table2), and OpenSora with rflow-30 (Table3).

### D.1.4 DESIGN CHOICES FOR DYNAMIC ORDER SELECTION

We tried different design choices for Dynamic Order Selection (DOS) and found that minimizing standard deviation is a simple and effective method for avoiding both overfitting from high-order oscillations and underfitting from low-order rigidity. Different design choices for DOS (1) Polynomial weighting functions of $t$, with powers ranging from 0.2 to 1.0. (2)Mean of residuals $\epsilon_n^{(i)}(t)$ across a local window. (3) Standard deviation of residuals $\epsilon_n^{(i)}(t)$ across a local window. We evaluated all strategies under the setting of $k=6$, $o=5$ and report FID-5k on DiT/XL-2.

The results are as follows: (1)Performance varies with exponent choice, as shown in Figure 10. (2)Mean-based selection yields FID = 13.89.(3)Standard deviation minimization yields FID = 9.87. Given its simplicity and superior empirical performance, we ultimately adopt standard deviation minimization as our default strategy for selecting the optimal order in DOS.

### D.1.5 THE EFFECT OF CACHE GRANULARITY

To further analyze the impact of caching granularity, we compare our approach, which caches the entire block output, with a finer-grained caching strategy that stores attention and MLP outputs at each timestep. All other settings (stride $k$ and maximum interpolation order $o$) are kept identical for a fair comparison. As shown in Table 7, the performance differences between the two caching granularities are minimal across various configurations, confirming that the primary performance gains stem from the TraCache mechanism itself rather than the specific choice of caching granularity.

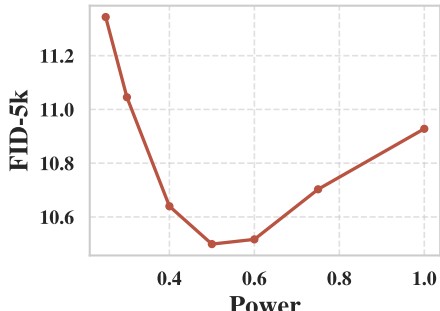

Figure 10: FID under different power exponents for $t^p$ weighting. The FID first decreases as the model better adapts to temporal variation, but increases beyond a certain point due to over-sensitivity to local noise.

Table 7: Comparison of different caching granularities for class-conditional image generation on ImageNet using DiT-XL/2. "attn/mlp" denotes caching attention and MLP outputs at each timestep, while "Ours" denotes caching the entire block output.

| Method | FID↓ | sFID↓ | IS↑ |
|---|---|---|---|
| attn/mlp (k=3,o=3) | 2.32 | 4.67 | 237.37 |
| Ours (k=3,o=3) | 2.34 | 4.54 | 237.51 |
| attn/mlp (k=4,o=4) | 2.55 | 4.78 | 234.82 |
| Ours (k=4,o=4) | 2.53 | 4.70 | 234.69 |
| attn/mlp (k=5,o=4) | 2.67 | 4.81 | 231.27 |
| Ours (k=5,o=4) | 2.69 | 4.63 | 230.67 |

## D.2 EXPERIMENT DETAILS

This section provides detailed hyperparameter configurations for the models used in our experiments, as introduced in Section 4.1: PixArt-$\alpha$ (Chen et al., 2023) (text-to-image generation), Open-Sora 1.2 (Zheng et al., 2024) (text-to-video generation), and DiT/XL-2 (Peebles & Xie, 2023) (class-conditional image generation).

**PixArt-$\alpha$.** FORA (Selvaraju et al., 2024) applies full attention and MLP computation every $k$ steps ($k$=2 for FORA[1], $k$=3 for FORA[2]). ToCa (Zou et al., 2024) computes attention and MLP for all tokens every $k$ steps, and performs partial updates for a ratio $r$ of tokens in the remaining steps, with $k$ and $r$ set according to the original paper.

**Open-Sora 1.2.** FORA is evaluated with $k$=2. For TeaCache (Liu et al., 2024a), we adopt the fast-version coefficient configuration from the official open-sourced implementation, and raise the threshold to 0.3 to realize an even faster variant.

## D.3 MORE EXPERIMENT RESULTS

### D.3.1 CLASS-CONDITIONAL IMAGE GENERATION WITH DDPM SAMPLER

In addition to the main experiments, we further evaluate TraCache under the DDPM sampler to validate its robustness and independence from the choice of sampling scheduler. Specifically, we replace the default DDIM sampler with DDPM while keeping all other experimental settings unchanged. As shown in Table 8, TraCache maintains a superior trade-off between efficiency and generation quality compared to other acceleration methods. This demonstrates that our approach generalizes well to different samplers and does not rely on sampler-specific characteristics.

Table 8: Quantitative results for class-conditional image generation with the DDPM sampler using DiT-XL/2 on ImageNet.

| Method | MACs(T) | FID↓ | sFID↓ | IS↑ |
|---|---|---|---|---|
| DiT-XL/2 | 57.20 | 2.28 | 4.52 | 276.49 |
| DDPM-125steps | 28.60 | 2.53 | 5.10 | 266.73 |
| FORA | 19.22 | 2.88 | 6.25 | 254.43 |
| ToCa | 19.79 | 2.63 | 5.88 | 258.54 |
| Ours | 19.34 | 2.48 | 5.037 | 267.78 |

### D.4 ADDITIONAL VISUAL RESULTS

Figure 11 compares our method with ToCa on PixArt-$\alpha$ at 512×512 resolution. Our method preserves more structural details and yields higher-fidelity textures compared to ToCa.

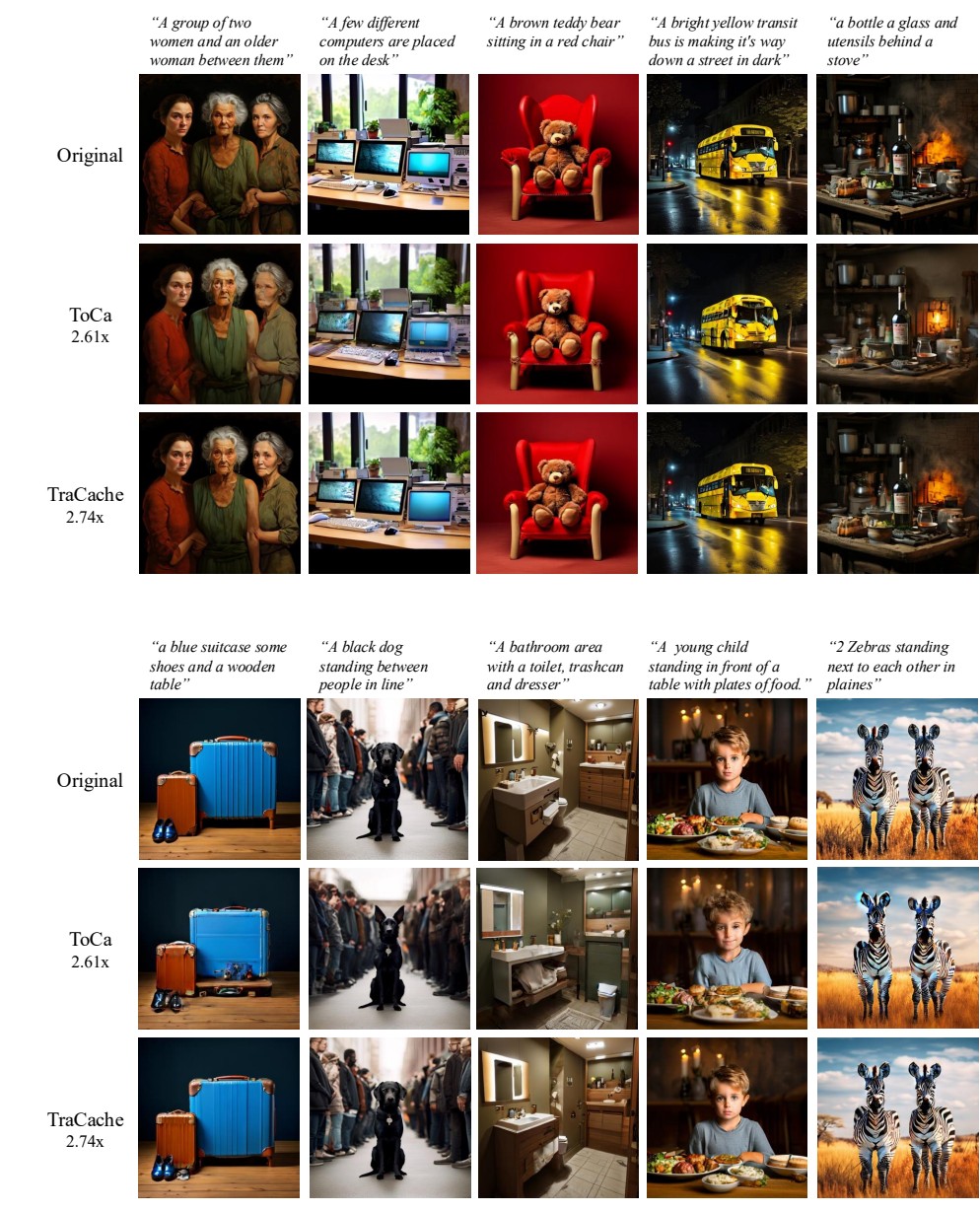

Figure 11: Visualization comparison of our method and ToCa on PixArt-$\alpha$ at 512×512 resolution.

We further present additional qualitative results on the text-to-image generation task using PixArt-$\alpha$ and the text-to-video generation task using Open-Sora, as illustrated in Figure 12 and Figure 13, respectively.

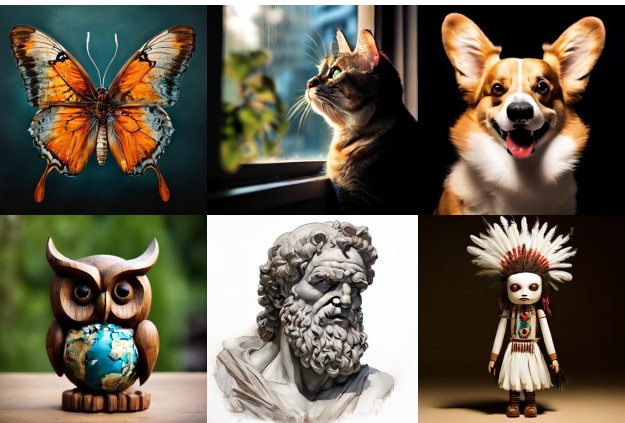

Figure 12: Qualitative results on the text-to-image generation task using PixArt-$\alpha$ under 2.74$\times$ acceleration (equivalent to 7 denoising steps).

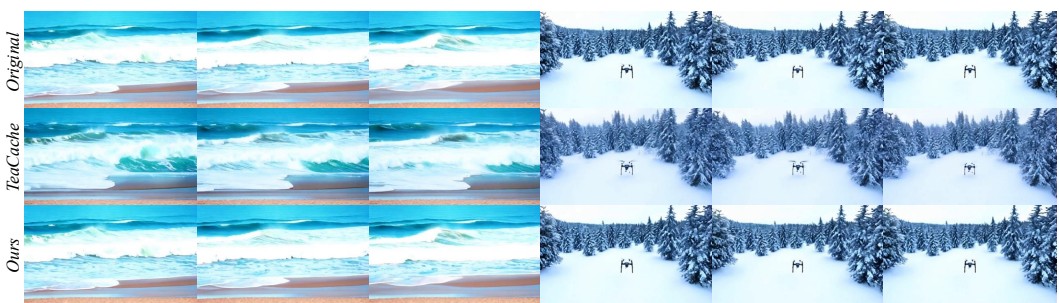

Figure 13: Qualitative results on the text-to-video generation task using Open-Sora under 3.74$\times$ acceleration (equivalent to 8 denoising steps).

