# OpenReview forum: "TraCache: Trajectory-Aware Feature Prediction for Training-Free Diffusion Transformer Acceleration"
_ICLR.cc/2026/Conference — Submitted to ICLR 2026_

### Official Review · Reviewer_CXfS · 2025-10-26

**Soundness:** 2
**Presentation:** 3
**Contribution:** 3
**Rating:** 6
**Confidence:** 4

**Summary:**

TraCache models the temporal evolution of features in the diffusion process to fit and extrapolate future representations, thereby maintaining generation quality under large-stride inference while significantly accelerating diffusion transformers.

**Strengths:**

1. This paper identifies the temporal smoothness of intermediate features in diffusion transformers and introduces TraCache, a trajectory-aware caching framework built upon this observation. It provides a principled alternative to conventional feature reuse, effectively mitigating the severe performance degradation observed in direct reuse methods as the temporal gap increases.

2. Extensive experiments across diverse generative tasks demonstrate the effectiveness of TraCache, showing that it consistently preserves superior generation quality under various acceleration settings compared to existing caching approaches.

3. The paper is clearly written and well-organized, making it easy to follow.

**Weaknesses:**

Video generation models inherently demand substantial GPU memory; however, the analysis of GPU memory usage in Table 4 is not sufficiently comprehensive or detailed in terms of both methodological explanation and model-level comparison.

**Questions:**

See Weakness

---

### Official Review · Reviewer_nLkR · 2025-10-29

**Soundness:** 2
**Presentation:** 2
**Contribution:** 2
**Rating:** 2
**Confidence:** 4

**Summary:**

This paper proposes TraCache, a training-free acceleration framework for Diffusion Transformers. The core idea is to model the temporal trajectory of intermediate features during denoising and replace direct cache reuse with local polynomial fitting and extrapolation to mitigate trajectory drift under large skipping strides. Extensive experiments on PixArt-α, Open-Sora, and DiT-XL/2 demonstrate that TraCache achieves high speedups while preserving near-original generation quality.

**Strengths:**

The paper introduces a trajectory-aware caching perspective, replaces naive reuse with polynomial-based prediction to reduce quality drop under large strides, employs a dynamic order selection mechanism that adapts to local smoothness, and achieves satisfactory acceleration results on both image and video generation tasks.

**Weaknesses:**

1. **Lack of comparison with recent prediction-based caching methods:** The paper does not cite or compare against TaylorSeer [1], AB-Cache [2], or HiCache [3], all of which also model feature trajectories using high-order Taylor expansion or multi-step ODE solvers. Without this, the novelty and technical distinction of TraCache remain unclear.

2. **Limited model coverage:** Experiments are restricted to previous diffusion-based DiTs (e.g., PixArt-α, Open-Sora) and do not include modern flow-matching models such as Stable Diffusion 3, FLUX, Qwen-Image, or HunyuanVideo. Since flow models exhibit smoother ODE trajectories, they are arguably better suited for trajectory prediction—omitting them weakens claims of general applicability.

3. **Ambiguities and potential numerical issues in formulation:** Equation 6 uses high-order Lagrange interpolation, which is prone to Runge’s phenomenon, yet no stability analysis is provided. The residual ϵn(i)(t) = ϕn(i)(t) − ϕn−1(i)(t) lacks definition for n=0. Moreover, Figure 8 shows severe FID degradation at high fixed orders—if TaylorSeer avoids this with Taylor expansion, it may indicate a fundamental limitation in the interpolation strategy.

4. **Presentation issues:** Table 1 uses Chinese quotation marks for “Res.”; Figures 8 and 9 have unreadably small fonts, harming readability.

[1] From Reusing to Forecasting: Accelerating Diffusion Models with TaylorSeers. ICCV 2025

[2] AB-Cache: Training-Free Acceleration of Diffusion Models via Adams-Bashforth Cached Feature Reuse. arXiv 2025

[3] HiCache: Training-free Acceleration of Diffusion Models via Hermite Polynomial-based Feature Caching. arXiv 2025

**Questions:**

1. Please explain why TaylorSeer and related works were not compared, and clarify the core differences in caching targets and prediction mechanisms.

2. Can you provide results on flow-matching models (e.g., FLUX, SD3)? Are they more suitable due to smoother trajectories?

3. In Equation 7, is std(ϵn(i)(t)) computed over spatial dimensions or a temporal window? Please specify.

4. Does the performance drop in Figure 8 suggest Lagrange interpolation is inferior to Taylor expansion? Please provide a direct order-wise comparison with TaylorSeer.

---

### Official Review · Reviewer_5AG1 · 2025-10-30

**Soundness:** 4
**Presentation:** 3
**Contribution:** 3
**Rating:** 4
**Confidence:** 4

**Summary:**

This paper proposes TraCache, a training-free, trajectory-aware feature caching framework. Existing feature caching methods suffer from quality degradation under large temporal gaps due to feature space trajectory shifts. TraCache mitigates this by modeling feature evolution across timesteps: instead of direct feature reuse, it fits local feature trajectories via polynomial interpolation and extrapolates future features. An adaptive dynamic order selection mechanism optimizes interpolation complexity for heterogeneous trajectory segments.

**Strengths:**

+ Good Insight into Core Problem: Instead of focusing on "when to cache" and "what to cache", the paper conducts an in-depth analysis of feature trajectory shifts, which identifies the root cause of quality degradation in existing caching methods.
+ TraCache’s trajectory-aware prediction avoids redundant computations without modifying the base model. The framework requires no additional training to model shifted trajectories, making it easy to deploy on pre-trained diffusion transformers. The combination of polynomial fitting and dynamic order selection balances accuracy and stability, adapting to both smooth and rapid feature dynamics.
+ Strong Empirical Performance: Results across 2 generative tasks and multiple models show consistent improvements. The paper performs ablation studies (on order selection, interpolation order, and cache granularity) to explore each component in TraCache, confirming the robustness and generalizability of the approach.

**Weaknesses:**

- Lack of Comparison with Similar Papers: The paper "From Reusing to Forecasting: Accelerating Diffusion Models with TaylorSeers" also adopts a "cache-then-forecast" approach, and there are other papers in this research direction that focus on predicting features at future timesteps. However, this paper neither mentions these works nor conducts any comparison with them. This part is my major concern.
- Gaps in PC Values: As shown in Figure 4, during the sampling process, there is still a certain gap between the PC values of different orders and the original ones.
- Limited Variety of Experimental Models: Experiments are not conducted on more popular diffusion transformers for image generation (such as FLUX, Stable Diffusion 3, and Qwen-Image) nor diffusion transformers for video generation (such as Hunyuan Video and Wan).
- Insufficient Exploration of Scalability to Higher Skip Rates: The paper does not conduct in-depth exploration of results with higher strides (e.g., stride > 5), nor does it clarify the stride size at which performance degrades significantly—this limits the upper bound of the method. Further exploration of trajectory modeling for ultra-aggressive acceleration could be valuable.

**Questions:**

Please refer to the Weaknesses section.

**Details Of Ethics Concerns:**

None.

---

### Official Review · Reviewer_CX33 · 2025-10-31

**Soundness:** 2
**Presentation:** 2
**Contribution:** 2
**Rating:** 2
**Confidence:** 5

**Summary:**

This paper introduces TraCache, a novel training-free framework designed to accelerate Diffusion Transformers. Instead of simply reusing old features, TraCache proposes to predict future features by modeling their evolution as a continuous trajectory. It uses a few historical feature outputs to fit a local polynomial function and then extrapolates this function to estimate the features for the current, skipped timesteps.

**Strengths:**

1. The paper is well-written and easy to follow
2. The theory proposed in this paper has solid mathematical foundations
3. The proposed method is simple and intuitive, and experiments have demonstrated its effectiveness.

**Weaknesses:**

1. The comparison of TraCache's VRAM usage relative to other methods is insufficient. Could you directly provide the peak VRAM usage of other methods in Table 2 and Table 3?
2. TraCache and TalorSeers (link: https://arxiv.org/pdf/2503.06923, accepted by ICCV25) are highly similar in their problem statement, core motivation, and overall structure. Additionally, given the same data points and equidistant timesteps, the polynomial fits obtained via Lagrange interpolation and the Taylor series expansion of derivatives approximated by finite differences should be identical. So can we conclude that these two papers share a high degree of fundamental similarity? Please provide a detailed explanation of your originality.
3. The additional computational overhead introduced by Dynamic Order Selection requires discussion.

**Questions:**

1. Is there a relationship between TraCache's trajectory prediction accuracy and the size of DiT parameters? For larger models, do we typically need higher polynomial orders?
2. Is this method robust to other model architectures, such as U-Net-based denoising models?

---

### Meta-Review · Area_Chair_QfmN · 2026-01-07

**Summary:**

This paper presents a method for accelerating DiTs by modeling the and extrapolating the temporal trajectory of intermediate features.
The submission received mostly negative reviews.
The reviewers mainly recognize the motivation and insights.
The main concerns from the reviewers were the limited evaluation (CX33, 5AG1, nLkR), limited model variety (5AG1 and nLkR), and lack of resource analysis (CX33 and CXfS).
The authors did not provide a rebuttal.
The AC agrees with the reviewers that the lack of experiments and comparisons does not justify for sufficient contributions.
As such, the AC recommends rejection.

**Reviewer Concerns:**

All reviewers' concerns remain outstanding.

**Reviewer Scores:**

I think all reviewers would keep their original ratings.

---

### Decision · Program_Chairs · 2026-01-26

Reject